# Design of Experiment Approach to Optimize Hydrophobic Fabric Treatments

**DOI:** 10.3390/polym12092131

**Published:** 2020-09-18

**Authors:** Iva Rezić, Ana Kiš

**Affiliations:** 1Department of Applied Chemistry, Faculty of Textile Technology, University of Zagreb, 10000 Zagreb, Croatia; 2Textile Company, Čateks, d.d. Ul. Zrinsko Frankopanska 25, 40000 Čakovec, Croatia; a.kis@cateks.hr

**Keywords:** polymer surface modification, hydrophobic properties, optimization, mathematical modeling, hydrophobicity

## Abstract

Polymer materials can be functionalized with different surface treatments. By applying nanoparticles in coating, excellent antimicrobial properties are achieved. In addition, antimicrobial properties are enhanced by hydrophobic surface modification. Therefore, the goal of this work was to modify the process parameters to achieve excellent hydrophobicity of polymer surfaces. For this purpose, a Design of Experiment (DoE) statistical methodology was used to model and optimize the process through six processing parameters. In order to obtain the optimum and to study the interaction between parameters, response surface methodology coupled with a center composite design was applied. The ANNOVA test was significant for all variables. The results of the influence of process parameters showed that, by increasing the pressure, concentration of hydrophobic compounds and dye concentration, water vapor permeability was enhanced, while by decreasing weight, its efficiency was enhanced. Moreover, the increase in the temperature enhanced water vapor permeability but decreased the resistance to water wetting. An optimal process with ecologically favorable 6C fluorocarbon (68.802 g/L) surpassed all preliminary test results for 21.15%. The optimal process contained the following parameters: 154.3 °C, 1.05 bar, 56.07 g/L dye, 220 g/m^2^ fabric. Therefore, it is shown that DoE is an excellent tool for optimization of the parameters used in polymer surface functionalization.

## 1. Introduction

Hydrophobic finishing of textile materials is one of the most important processes in the textile industry. Excellent finished compounds repel water, oil and dry dirt which is extremely important in clothing, sports, military, medical and technical textiles [1]. Polymer materials can be functionalized to become active against resistant microorganisms with different surface treatments. For example, by applying nanoparticles inside the surface coating, excellent antimicrobial properties are achieved. In addition, antimicrobial properties are enhanced by further surface modification that enables excellent hydrophobic properties.

Historical usage of water repellant materials covers a broad area of wax and resin compounds that can be easily washed out [2]. Today perfluoroalkyl compounds (PFAS’s) are widely used in almost all staining repellant finishes since those are the only chemicals capable of simultaneously repelling water, oil, dirt and all other staining compounds [3]. In addition, by enhancing the repellant properties, other fabric properties show much better performances, including increased resistance to acids, bases and other chemicals, more rapid drying, and better durable press properties [4].

Fluorocarbons are synthetically produced organic chemicals in which all hydrogen atoms are replaced by fluorine containing a perfluoroalkyl residue (Figure 1) [1]. Those chemicals exhibit a significant reduction in surface tension (due to their incompatibility with water and oil), and outstanding stability (both chemical and thermal) [3].

The mechanism of repellency includes reducing the free energy at the material surface [5]. Therefore, if the adhesive forces between material and drops of liquid on the material are greater than the internal cohesive interactions within the liquid, the drops will spread. In contrast, in cases when adhesive interactions between the material and the liquid are smaller than the internal cohesive interactions within the liquid, the liquid drops will not spread [6].

There are many parameters which influence the surface tension and thereby repellency, and one of them is the length of the chain [3]. The literature data show that the critical surface tension decreases rapidly as the chain length increases from one to eight, and after that decreases [7]. However, fluorocarbons with 8 C atoms (“C8”) have a negative impact on the environment and human health so, from September 2020, the Croatian industry will replace all ecological non favorable substances with C6 fluorocarbons which are allowed by strict regulations. This puts extra requirements on industrial managers and process engineers that need to meet demands of rigorous ecological standards, as well as customer requirements for super hydrophobic garments.

Different treatments are developed using cost effective nanotechnology-based repellents, but new formulae need improvement since current products have a negative impact on comfort in hot and humid environments [8]. Therefore, in order to enhance the repellency and other protective properties of garments, many efforts are being conducted in different processing areas: fabric weave technology [9], coating technology, surface nanocomposite modifications with resins and nanoparticles [10] and double-sided knitting consisting of two different kinds of fibers [11]. Moreover, achieving waterproof breathable fabrics [12], super-hydrophobic and multi-responsive fabric composite with excellent electro-photo-thermal effects [13] with applications not only to textiles but also in composite insulators [14] and producing intelligent electronic clothing systems [15], is under investigation. Recent developments also cover the area of medical materials with antibacterial capacity [16,17], as well as methods for the incorporation of membranes, coatings, fabrics, lining material and other vital parameters for designing breathable garments [18].

Many parameters influence the wear comfort of materials i.e., porosity and strength of the micro-porous layer on textile material, such as: crystallinity, temperature and rate of stretching, pressure, temperature and duration of the heat treatment, the choice of reagents and others [1]. In order to find optimal conditions for the multi-parameter process, Design of Experiment (DoE) statistical methodology is often used [19,20,21,22,23]. For example, Moussa et al. used a factorial experimental design for optimization of the waterproof breathable property of samples [11] and optimized vital parameters. Coronado et al. used the mixture design of experiments to assess the environmental impact of clay-based structural ceramics containing hazardous metals As, Ba, Cd, Cr, Cu, Mo, Ni, Pb, and Zn [19], and Waha et al. optimized the process of biodegradation by the Taguchi design of experiment [20].

Long chain fluoro-chemicals with at least eight perfluorinated carbon atoms are synthetic compounds that have been used since the 1940s in a wide variety of consumer and industrial products such are firefighting foams, surfaces, and food contact paper. Due to their widespread usage, today they can be detected in the environment, as well as in humans. Unfortunately, recent evidence shows that continued exposure to above specific levels of certain long chain fluorochemicals may lead to adverse health effects [24]. Recent investigations have proved that PFAS’s are associated with reproductive toxicity, reduced growth metrics in newborns and elevated cholesterol levels in humans. Moreover, PFAS are highly persistent and resistant to degradation and are therefore a serious global concern.

According to some predictions, by 2050, more people could die from the infections caused by antibiotic-resistant bacteria than from cancer. Only in Europe 25,000 deaths per year and costs over EUR 1.5 billion are associated with resistant microorganisms. Especially dangerous infections are bacteria such as *Staphylococcus aureus*, *Methicillin-resistant Staphylococcus aureus* (*MRSA*)*,* known as “super-bacteria” or “golden staphylococci”, which is increasingly difficult to cope with due to its’ resistance to a wide range of antibiotic-based penicillin drugs (β-lactam antibiotics such as methicillin, dicloxacillin, nafcylin and oxacillin). *Staphylococcus aureus* is a member of the *Staphylococcaceae* family of gram-positive bacteria of spherical forms and is one of the most significant pathogens in the world. It is an infection with a frequency ranging from 20 to 50 cases per 100,000 inhabitants per year, with 10% to 30% of infections ending with a deadly outcome. This number is greater than the sum of the deaths caused by AIDS, tuberculosis and viral hepatitis combined.

Therefore, the aim of this research was to develop a hydrophobic polymer surface that can be further used as potential antimicrobial material. The idea was to achieve excellent results of hydrophobic properties through the optimization of textile functionalization process parameters. Optimization was performed through six process parameters using Design of Experiment. The goal of optimization was to find optimal conditions of the water vapor permeability and the resistance to water wetting. The analysis of water vapor permeability is related to the level of comfort at low physical activity but does not give the information about condensation on a textile surface.

## 2. Materials and Methods

### 2.1. Samples

In this research three different samples of textile material intended to be used as military garments were investigated. Samples were made of cotton and polyamide yarns with weights of 190 g/m^2^, 220 g/m^2^ and 240 g/m^2^. The mechanical properties of samples are presented in Table 1.

### 2.2. Reagents and Chemicals

The hydrophobic compounds tested were a fluorocarbon agent based on C8 fluorochemicals, Sevophob HFK–F, (fluorocarbon resin which is used for permanent water, oil and dirt-repellent finishing, producer: Textil Color, Sevelen, Switzerland), and a fluorocarbon agent based on C6 fluorochemicals, Tubiguard SCS-F, (low viscous liquid dispersion, producer: CHT) in concentrations from 35 to 70 g/L. The dye used during the textile finishing treatment was Bezathren Navy Blue GN vat dye (producer: CHT, Montlingen, Switzerland) since it offers an outstanding light, wet and chlorine fastness level like no other dye class on cellulose fibers. It was applied in the form of a micro-disperse powder which is easily dispersed and used in textile modification.

### 2.3. Textile Finishing Treatment

The textile finishing treatment used for the modification of the material into a hydrophobic textile was pad impregnation of the dyeing of fabric and for the application of finishing chemicals. It had several steps: (1) pre-treatment for removal of waste, oil, dry matter and other impurities; (2) dyeing; and (3) finishing with two different types of fluorocarbon agents to achieve water and oil repellency. The second and the third steps (dyeing and finishing experiments) were performed on a laboratory scale using laboratory machinery for padding in which the fabric passes into a solution of dye and chemicals under a submerged roller, and then it goes out of the bath is squeezed to remove excess solution by pressure occurring between two cylinders. The objective of this process is to mechanically impregnate the fabric with the solution or dispersion of chemicals.

### 2.4. Determination of Water Vapor Permeability

Moisture transport through textiles is the factor which influences thermological and physiological comfort of the material. The moisture is transferred through a material in the form of vapors or liquids.

Water vapor permeability, (WVP) is expressed as the time rate of water vapor transmission through a unit area induced by unit vapor pressure difference between two specific surfaces, under specified temperature and humidity conditions [25]. It is calculated as:(1)$$WVP=\;\frac{G}{t}\times A\times \Delta p$$
where ∆*p*, (*Pa*) is the difference of partial pressure between two sides of material, *G*, (g)—weight change, *t*, (*h*)—time during which *G* occurred, and *A*, (m^2^)—test area.

In the ASTM E96 standard procedure the test cup is filled with the distilled water and the circular sample is tightly covered onto the cup. The cup prepared for testing is placed in a controlled environment at an ambient temperature of 23.0 °C, with a relative humidity of 100% inside the cup and 50% outside the cup. Due to the forces of the differences in concentration and pressure, vapor diffusion occurs through a textile from the cup in this environment [25].

After this treatment, the resistance of the hydrophobic modification was tested in a 5-cycle process washing according to the norm EN ISO 6330:2012 Textiles—Domestic washing and drying procedures for textile testing, testing procedures at 60 °C. In this work, the WVP was tested according to the ASTM E96 standard procedure in which the vapor passes from inside of the cup to the outside of the cup (Figure 2).

The drawback of this methodology is the fact that the permeability that is measured by this method depends not only on properties of a material, but also on the thickness of the air layer near the surface of the textile material. Therefore, the resistance to surface wetting was measured in another set of experiments (Section 2.5).

### 2.5. Resistance to Surface Wetting

A resistance to surface wetting test was performed according to the ISO 4920:2012 standard procedure [26]. This method specifies spray test conditions for determining the resistance of fabric to surface wetting by water in the following manner: a specified volume of distilled water is sprayed onto a test specimen that has been mounted on a ring and placed at an angle of 45°. It is placed in a way that the centre of the specimen is at a specified distance below the spray nozzle. The spray rating is determined by comparing the appearance of the specimen with descriptive standards.

### 2.6. Mathematical and Statistical Procedures

Section 2.1, Section 2.2, Section 2.3, Section 2.4 and Section 2.5. were statistically analyzed using Design Expert Stat Ease software 9.1 (Minneapolis, MN, USA). A central composite design was chosen for the modeling of six process parameters with replicates, so the total number of experiments was 42. The optimal conditions for six independent variables, i.e., fabric weight, dye concentration, concentration of hydrophobic compounds Sevophob HFK–F and Tubiquard SCS–F, temperature, pressure and the type of the hydrophobic compound, were obtained using algorithms with the Design Expert State Ease version 9.1 software (Minneapolis, MN, USA). The critical surface tension of hydrophobic compounds decreases rapidly as the chain length increases from 1 to 8, and after that decreases [7]. In majority of industrial processes fluorocarbons with 8 C atoms were frequently used due to their extraordinary hydrophobic properties. However, due to their extremely negative impact on the environment and human health, their usage is prohibited so the industry needs to find a suitable replacement. Therefore, in this work the efficiency of different treatments using C 6 and C 8 fluorocarbons was investigated and optimized through six different industrial process parameters.

This statistical program DoE searches for an optimal combination of factor levels that simultaneously satisfies the requirements placed on each of the responses and factors.

## 3. Results

This section is be divided by subheadings. It should provide a concise and precise description of the experimental results, their interpretation as well as the experimental conclusions that can be drawn. Sevophob HFK–F and Tubiquard SCS–F are fluoro-based finishing compounds that were investigated in this research as efficient hydrophobic reagents that enable durable and high-performing finishing of military garments. Sevophob HFK-F is much a more efficient reagent, but contains eight C atoms in its structure, while Tubiquard SCS-F has only six C atoms.

In order to achieve equally satisfying results with an ecologically acceptable reagent, multi-parameter optimization using Design of Experiment was performed. Therefore, in this work six process parameters were investigated, namely: the weight of the military fabric (ranging from 190 to 240 g/m^2^), concentration of the dye (10 to 60 g/L), concentration of hydrophobic compounds (35 to 70 g/L), type of hydrophobic compounds (ranging from six to eight carbon atoms), temperature (from 150 to 170 °C) and pressure (from 1.0 to 2.0 bar). In total, six input process variables were varied for optimization purposes in 42 preliminary experiments, with the goal to optimize the hydrophobic properties of the garments, while preserving the wear comfort of the materials.

### 3.1. Optimization by Design of Experiments

A traditional optimization protocol monitoring one parameter at the time cannot provide information on interactions between the process parameters and their outcome for the industrial process. Design of experiment (DoE) statistical methodology offers the answer to this problem and the possibility to study the effects of process variables and their responses within the minimal number of preliminary experiments [27]. 

Using DoE based on the response surface methodology (RSM) within the State Ease software, the aggregate mix proportions were derived, and the total number of experiments was drastically reduced. In order to examine whether there is a relationship between the selected parameters and the response variables investigated, the collected data needed to be analyzed in a statistical manner using regression according to Equation (2):(2)$$x_{i}\;=\;\left(X_{i}-\;X_{i}^{x}\right)/\Delta X_{i}$$
where *x_i_* is the coded value of the independent variable *i*, *X_i_* the natural value of the variable *i*, *X_i_^x^* the natural value of the variable *i* in the central point and ∆*X_i_* is the value of the step change.

The response *y* (estimation of the coefficients of a quadratic model) is represented by Equation (4), within the central composite design:(3)$$y=\beta_{0}+\overset{k}{\underset{i=1}{\sum }}\;\beta_{i}\;x_{i}{}^{\;}+\overset{k}{\underset{i=1}{\sum }}\;\beta_{ii}x_{i}^{2}+\overset{\;}{\underset{i<j}{\sum }}\;\beta_{ii}\;x_{i}\;x_{j}+\varepsilon$$
where, *y* is the measured response, *β*_0_ is the intercept term, *β_i_*, *β_ii_* and *β_ij_* are the measures of the effects of variables *x_i_*, *x_i_x_j_* and $x_{i}^{2}$, respectively. The central composite design was selected since this is one of the most efficient and popular classes of second order designs recommended in the literature [19]. A very important optimization step is the selection of parameters and their ranges to be studied, which was done based on previous analysis. The factors were studied in several levels, from low to high, as is presented in Table 2.

As can be seen from the Table 2, three experiments (runs 3 and 39, 4 and 28, 11 and 12) were repeated to check the reproducibility and to estimate an experimental error. All three responses gave a reproducible result where the deviation of each run was found within an experimental error (i.e., ±0.075% for water vapor permeability, and ±0.053 for resistance to water wetting). The worst result in the preliminary tests was recorded in the eighth experiment (with 1196 g/m^2^ for water vapor permeability), for the hydrophobic compound with six C atoms.

The center composite design matrix of 42 experiments covering full design of three level factors was used to build a quadratic model of the experimental data of the observed responses. Standard error of design is shown in Figure 3. It reports the standard error of the predicted mean in a way in which larger standard error means less reliable the estimates.

### 3.2. ANNOVA Report

The analysis of variance (ANOVA) was carried out to establish the significance of different parameters for the quadratic model. The quadratic model for the water vapor permeability in terms of coded factors is presented in Equation (4) as:(4)$$\begin{array}{ll} \mathrm{Water}\;\mathrm{Repellency}\; & =\;87.41\;-\;1.19\;\times \;A\;+\;0.12\;\times \;B\;+\;0.49\;\times \;C\;+\;4.02\;\times \;D\\ & -\;0.20\;\times \;E\;-\;9.87\;\times \;F\;+\;0.15\;\times \;\mathrm{AB}\;+\;0.33\;\times \;\mathrm{AC}\;+\;3.81\\ & \times \;\mathrm{AD}\;-\;3.96\;\times \;\mathrm{AE}\;-\;2.23\;\times \;\mathrm{AF}\;-\;1.64\;\times \;\mathrm{BC}\;-\;6.21\;\times \;\mathrm{BD}\\ & -\;0.012\;\times \;\mathrm{BE}\;+\;3.56\;\times \;\mathrm{BF}\;-\;0.34\;\times \;\mathrm{CD}\;-\;1.21\;\times \;\mathrm{CE}\\ & +\;0.048\;\times \;\mathrm{CF}\;-\;0.91\;\times \;\mathrm{DE}\;+\;2.89\;\times \;\mathrm{DF}\;+\;0.49\;\times \;\mathrm{EF}\\ & +\;3.07\;\times \;B^{2}\;-\;4.85\;\times \;C^{2}\;-\;1.14\;\times \;D^{2}\;+\;0.64\;\times \;E^{2} \end{array}$$
where coded parameters are: *A* Fabric weight, *B* Dye concentration, *C* Hydrofobe concentration. *D* Temperature and *E* the Pressure. The significance of parameters is confirmed if their *p*-values are below value of 0.05 (which is the significance limit).

In this model, all parameters were significant having *p* values of 0.0020, 0.0109, 0.0005, 0.0220, 0.0010 and 0.0001 for fabric weight, dye concentration, hydrophobic compound concentration, temperature, and pressure, respectively. The model *F*-value of 4.22 implies that the model is significant according to a 95% level of confidence.

Values of “*p > F*” less than 0.0500 indicate that the model terms are significant and in this work the *p*-value was 0.0022. The lack of fit was calculated from the experimental error (pure error) and residuals. The lack of fit is the ratio between the residuals and pure error. “Lack of fit *F*-value” of 10.13 with its *p*-value of 0.0096 implies the lack of fit was significant. Therefore, the suggested model for water vapor permeability presented in Equation (4) is valid for the present study.

### 3.3. Significant Graphics

Figure 4 compares experimental values with the predicted model value obtained from Equation (4). The value of the correlation coefficient, R^2^ was found to be high (i.e., close to unity) which confirms the accuracy of the model [20,21].

The Durbin–Watson statistic presented in the Figure 5 shows that no correlation can be observed between the experimental and calculated values so there is no evidence of correlation in the residuals’ series, and therefore there is no accumulation of experimental error [19].

### 3.4. Selectivity

The model from Equation (4) is presented graphically in Figure 6, and the study of the effects of process variables on the response water vapor permeability is presented in Figure 7.

As can be seen in Figure 7, six parameters mutually influence the response water vapor permeability, but also show interactions and synergistic effects on the output variable.

### 3.5. Optimization Using Response Surface Methodology

Resistance to water wetting occurred both when using Sevophob HFK–F and Tubiquard SCS–F through lowering the surface energy of the fabric, so that water did not wet out the garment. Although similar finishes could be achieved with some other types of finishers (such are waxes, oils and silicones), those other reagents are not repellent to oil, sand or lotion compounds.

For this reason, the fluorocarbons such are Sevophob HFK–F and Tubiquard SCS–F, were used as the most effective compounds for repelling both oil and water, which is a very important parameter in finishing military garments. However, Sevophob HFK-F contains eight C atoms so the optimal process in which Turbiquard can be used, but with equal efficiency needed to be developed.

Zahid et al. reported that among materials with extra developed hydrophobic properties, textile materials are come in contact with the human skin most frequently. Therefore, the authors have emphasized that textile treatments for water or oil repellency should be non-toxic, biocompatible, and comply with stringent health standards [28]. Moreover, due to the large quantities of water, chemicals and reagents used worldwide in the textile industry, treatments should be scalable, sustainable, and eco-friendly. Due to this awareness, new eco-friendly processes are being developed and adopted.

Moreover, the review article of Zahid et al. reported that although fluorinated polymers with C8 chemistry are the best performing materials to render textiles water or oil repellent, they pose substantial health and environmental problems and are being banned. Therefore C8-free vehicles for non-wettable treatment formulations are probably the only ones that can offer important commercialization prospects. In addition, their review article indicated promising future strategies and new materials that can transform the process for non-wettable textiles into an all-sustainable technology [28].

In this work, the model Equation (4) was used to find the manner in which the parameters need to be varied in order to achieve the optimal solution for obtaining hydrophobic properties with less harmful reagents that contain six C atoms. The corresponding optimized surface response of the quadratic model is shown in Figure 8.

The surface contour plots of parameter interactions between industrial process variables are elliptical, and the central point is the point in which the slope of the contour in all directions is equal to zero. The minimal predicted response yield is indicated by the surface confined in the smallest curve of the contour diagram, so the optimal maximized solutions are outside this area, as is shown in Figure 8.

The analysis of the results presented in Figure 8 shows the significant influence of the process parameters on the results. As can be seen from this Figure, by increasing the concentration of hydrophobic compounds, concentration of the dye and pressure, leads to the enhancement of water vapor permeability. In contrast, by decreasing the garment weight, its efficiency was enhanced. Moreover, the increase in temperature enhanced water vapor permeability but decreased the resistance to water wetting. Those results can be explained by the fact that, at high concentration of reagents and rigorous process variables, optimal coating finishing will occur.

Statistical analysis of this model results in the optimal solution for maximized water vapor permeability. Therefore, the additional experiments were carried out at optimized conditions and the results of the responses predicted by DoE were compared to experimental results as is shown in Table 3. Table 3 presents verification experiments of the model at three optimal solutions for three different kinds of textile fabric (weights 190, 220 and 240 g/m^2^, respectively).

As can be seen from this table, two optimal solutions were obtained with eight C atoms, but another two were obtained with six C atoms, and this result surpassed all preliminary tests for both six C atoms and eight C atoms by 21.15%. However, when this optimal result of 2620 g/m^2^ is compared with the statistical mean value obtained only for six C atoms (1726.86 g/m^2^), the global optimum was 51.72% better. This means that the industrial process can be performed under the presented variable parameters achieving satisfying results for both ecologically and economically.

Excellent resistance to water wetting remained even after five washing cycles, which is an excellent result. Without optimization, the resistance to water wetting effects would drop to 50–70%, and our optimized process resulted with a modification that preserved resistance to water wetting (at 90–100%).

Figure 9 shows the desirability for each factor and all responses individually for the selected optimal solution. As can be seen from the Table 3, for the optimal result, there was an error of ±1.96% for resistance to water wetting and error of ±2.28% for water vapor permeability with a 95% confidence level (i.e., 94 ± 0.52%).

The verified optimized solution of the model responses of water vapor permeability and resistance to water wetting are shown in Figure 10.

Based on the optimization model shown in Figure 10, the optimum of responses (water vapor permeability and resistance to water wetting) could be predicted and verified by comparing the predicted result with the new experimental output.

An optimal process with an ecologically favorable fluorocarbon with six C atoms (68.802 g/L, Tubiquard) surpassed all test results by 21.15%. It was found at 154.3 °C, 1.05 bar, 56.07 g/L dye, 220 g/m^2^ fabric, which surpassed the results of statistical mean for six C atoms by 51.72%. Moreover, the best solution obtained with hydrophobic compound of eight C atoms was 2609 g/m^2^. It was found at 152.5 °C, 1.9 bar, 38.815 g/L of Sevophob HFK-F, 11.1 g/L dye on 240 g/m^2^ fabric, and surpassed the statistical mean of all test results with eight C atoms (1619.2 g/m^2^) by 61.12%. From this it can be concluded that Design of Experiment is an excellent tool for optimization of multi-parameter complex systems because with high numbers of factors only a fraction of the experiments need to be completed to efficiently estimate the main effects and parameter interactions [27].

## 4. Conclusions

The presented work shows that Design of Experiment is a convivial tool for optimization of modification of the surface since the influence of the various process parameters can be easily investigated. In this work, fabric weight, dye concentration, hydrophobic reagent concentration, temperature, pressure, and the type of the hydrophobic compound, have been investigated to determine the optimal conditions. Validity of the regression equation has been controlled by a statistical approach, and the results have revealed following conclusions:-Sevophob HFK–F and Tubiquard SCS–F are efficient hydrophobic reagents, with Sevophob HFK–F being much more efficient but ecologically unacceptable.-Central composite design of experiments was used based on 42 preliminary experiments in order to investigate six process parameters within laboratory scale optimization, to find an alternative solution using Tubiquard SCS-F with six C atoms.-An Ecologically favorable optimum using a fluorocarbon with six C atoms (68.802 g/L, Tubiquard) surpassed all the test results by 21.15%. It was found at 154.3 °C, 1.05 bar, 56.07 g/L dye, 220 g/m^2^ fabric.

From this it can be concluded that efficient water repellency can be achieved using six C fluorocarbons while ecologically harmful and forbidden hydrophobic compounds with eight C atoms can be successfully replaced by reagents acceptable for the industry purposes.

## Figures and Tables

**Figure 1 polymers-12-02131-f001:**
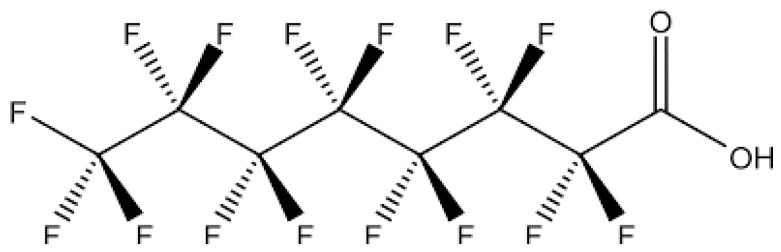
Perfluorooctane alcohol: F_3_C–CF_2_–CF_2_–CF_2_–CF_2_–CF_2_–CF_2_–CF_2_–CF_2_–CH_2_–CH_2_–OH [3].

**Figure 2 polymers-12-02131-f002:**
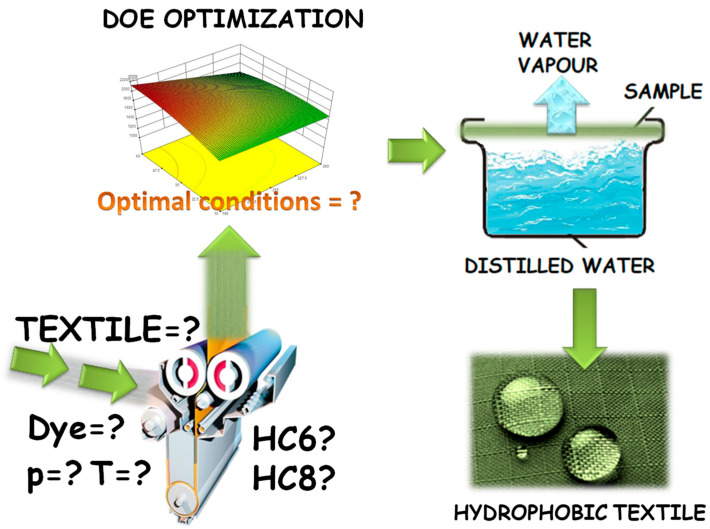
Schematic overview of process for optimizing the highest water vapor permeability by the ASTM E96 method.

**Figure 3 polymers-12-02131-f003:**
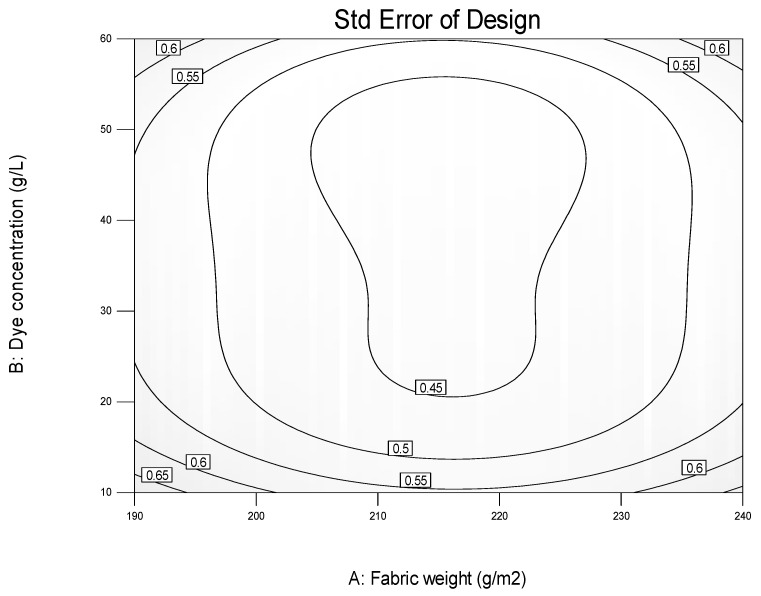
Standard error of design.

**Figure 4 polymers-12-02131-f004:**
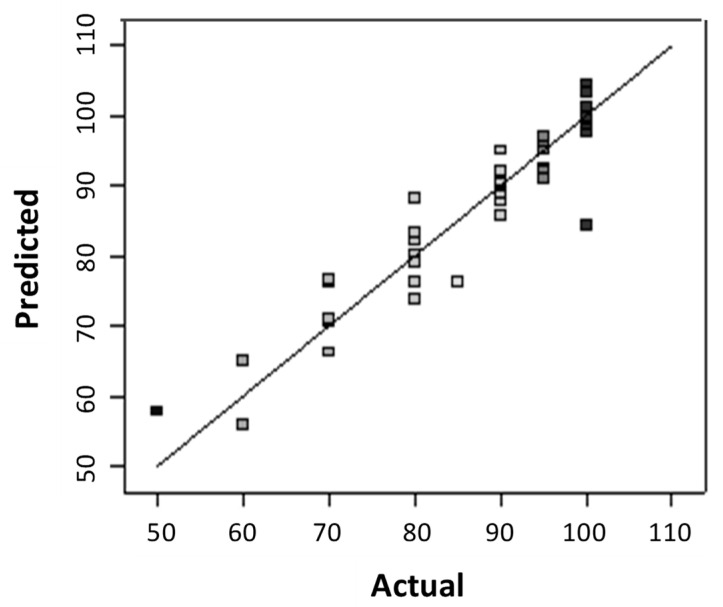
Plot of actual value and the value predicted from the Equation (4).

**Figure 5 polymers-12-02131-f005:**
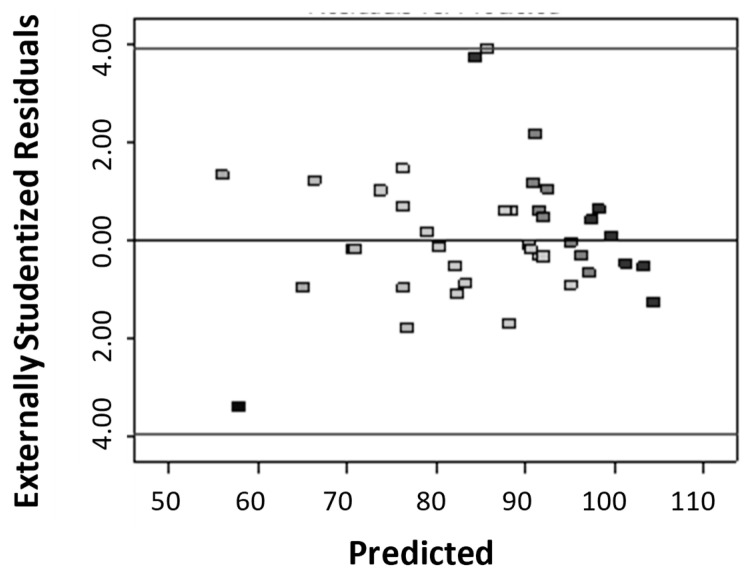
Residual correlation: q(experimental)–q(calculated).

**Figure 6 polymers-12-02131-f006:**
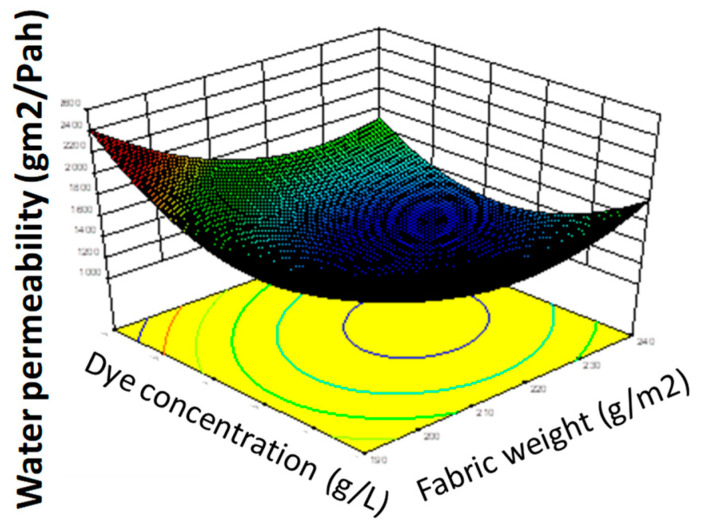
Graphical presentation of the model equation, effect of fabric weight and the dye concentration on the response parameter water vapor permeability.

**Figure 7 polymers-12-02131-f007:**
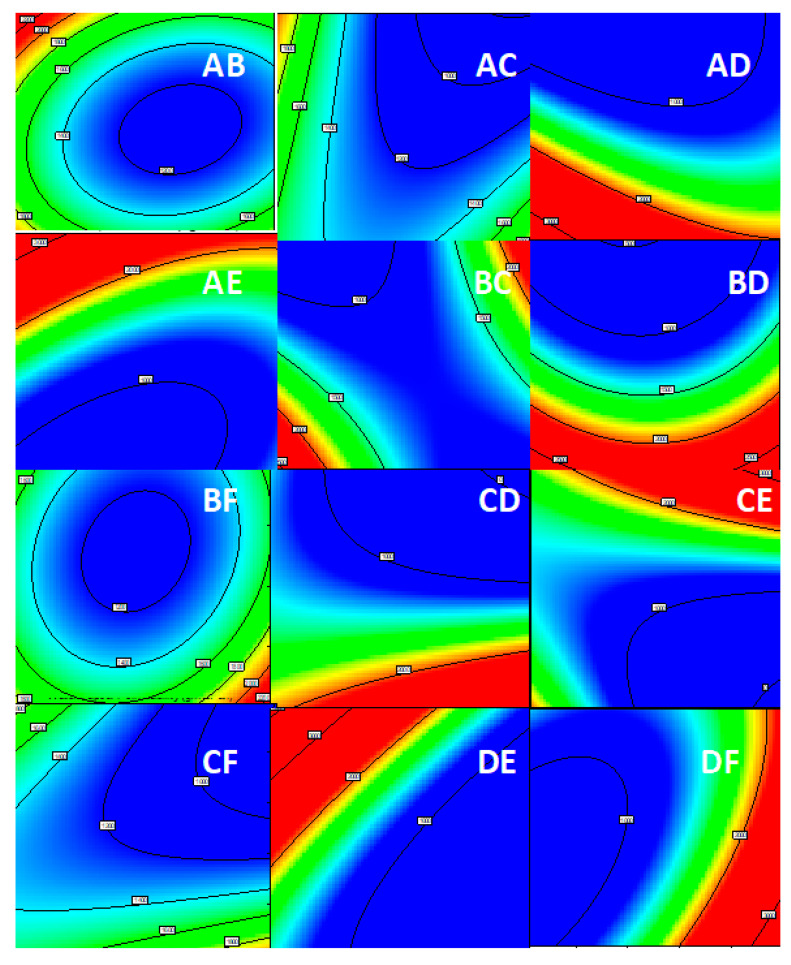
Effects of industrial process parameters on response water vapor permeability: (**A**—fabric weight, **B**—dye concentration, **C**—hydrophobic concentration, **D**—temperature, **E**—pressure and **F**—the type of the hydrophobic compound). Interactions shown are: (**AB**,**AC**,**AD**,**AE**,**BC**,**BD**,**BF**,**CD**,**CE**,**CF**,**DE**,**DF**).

**Figure 8 polymers-12-02131-f008:**
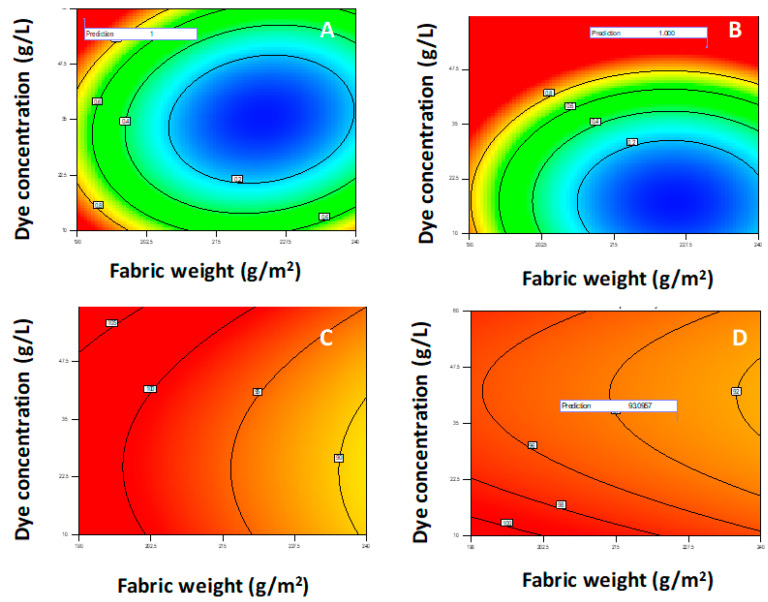
(**A**–**D**): Surface contour plots for desirability, water vapor permeability, and resistance to water wetting at optimal values of initial fabric weight (g/m^2^) and dye concentration (g/L) for different values of other parameters: (**A**) C = 51.386, D = 159.109, E = 1.504, F = Sevophob HFK-F; (**B**) C = 61.437, D = 156.768, E = 1.228, F = Tubiquard SCS-F; (**C**) C = 51.756, D = 150.309, E = 1.995, F = Sevophob HFK—F; (**D**) C = 38.136, D = 159.208, E = 1.852, F = Sevophob HFK—F, where the other parameters are C—hydrophobic concentration, D—temperature, E—pressure and F—the type of the hydrophobic compound.

**Figure 9 polymers-12-02131-f009:**
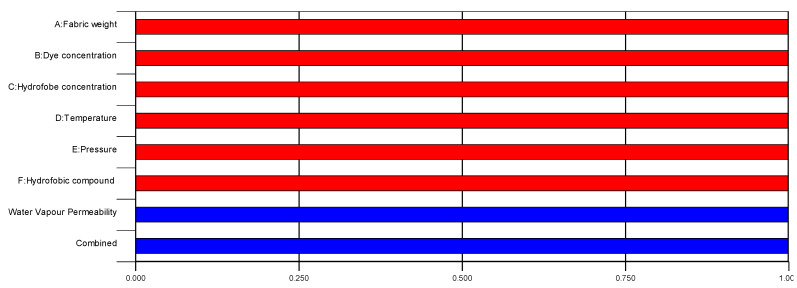
Desirability for each factor and each response individually for the optimal solution.

**Figure 10 polymers-12-02131-f010:**
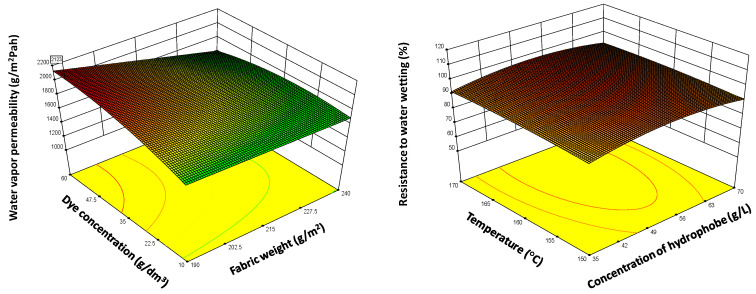
Response surface of the Central composite design, optimized response variables for: water vapor permeability and resistance to water wetting.

**Table 1 polymers-12-02131-t001:** Mechanical properties of investigated samples.

Weight, g/m^2^	Fabric Composition	Density, Thread per 1 cm		Yarn Count, Tex *		Construction of Fabric
		Warp	Weft	Warp	Weft	
190	50% cordura (PA 6.6. Type 420 HT dull)/50% cotton	35.8	19.5	14 × 2 tex	14 × 2 tex	Ripstop (square 7 ± 1 mm × 7 ± 1 mm)
220		35.8	20.5	14 × 2 tex	38.4 × 1 tex	
240		35.8	20.5	15 × 2 tex	50 × 1 tex	

** Tex* is a direct measure of linear density and represents grams per kilometre of the yarn.

**Table 2 polymers-12-02131-t002:** Experimental planning in Design of Experiment, central composite design, six parameters, 42 preliminary experiments, six input parameters and two output responses.

PARAMETERS							RESPONSES	
Random Run	Fabric Weight g/m^2^	Dye Concentration g/L	Concentration of Hydrophobic Compound FC, g/L	Temperature °C	Pressure Bar	Type of Hydrophobic Compound	WVP, g/m^2^, 24 h	Resistance to Water Wetting
1	240	60	70	170	1.66	Tubiquard SCS-F	1658	70
2	190	60	70	150	1	Tubiquard SCS-F	2090	80
3	220	40	35	161	1.85	Sevophob HFK-F	1646	90
4	190	59.2	47.6	159.8	1.39	Sevophob HFK-F	1824	90
5	240	10	62.1	166.5	1.43	Tubiquard SCS-F	1738	100
6	220	60	70	164	1.3	Sevophob HFK-F	1517	90
7	220	29	70	153	1.77	Tubiquard SCS-F	1963	60
8	220	40	41.8	160.7	1.2	Tubiquard SCS-F	1196	70
9	240	16.5	38.2	170	1.38	Tubiquard SCS-F	1617	80
10	190	22.8	35	150	2	Sevophob HFK-F	2000	95
11	190	13.8	60.2	157.4	1.66	Tubiquard SCS-F	1466	80
12	190	13.8	60.2	157.4	1.66	Tubiquard SCS-F	1390	80
13	220	52.5	35	170	1	Sevophob HFK-F	1765	95
14	240	35	50.9	159.9	2	Tubiquard SCS-F	1463	70
15	240	30.0	67.9	170	2	Tubiquard SCS-F	1522	80
16	240	10	70	164.6	2	Sevophob HFK-F	1696	100
17	220	49.8	61.3	150.3	1.05	Sevophob HFK-F	1926	100
18	190	12.8	70	150	1	Sevophob HFK-F	1770	95
19	190	13.5	70	170	1	Tubiquard SCS-F	1825	80
20	220	60	51.7	150	2	Tubiquard SCS-F	2007	80
21	220	10	54.5	158	1.25	Sevophob HFK-F	1247	100
22	240	35	68.0	166.0	1	Sevophob HFK-F	1720	100
23	240	60	35	150	1.59	Tubiquard SCS-F	1560	70
24	220	10	35	150	2	Tubiquard SCS-F	1296	60
25	240	60	42.7	170	2	Sevophob HFK-F	1362	90
26	240	60	70	150.5	2	Sevophob HFK-F	1558	90
27	240	26.9	54.3	150	1.52	Sevophob HFK-F	1653	90
28	190	59.2	47.6	159.8	1.39	Sevophob HFK-F	1647	95
29	240	10	70	150	1	Tubiquard SCS-F	1585	50
30	240	56.3	61.1	155.9	1.45	Tubiquard SCS-F	1796	85
31	240	27.0	54.3	150	1.52	Sevophob HFK-F	1995	95
32	240	10	35	157.5	1	Sevophob HFK-F	1834	100
33	220	60	35	150	1	Sevophob HFK-F	1720	95
34	190	60	35	169.5	2	Tubiquard SCS-F	1973	90
35	190	10	35	170	1.6	Sevophob HFK-F	1892	100
36	240	60	47.3	166	1	Tubiquard SCS-F	1560	90
37	220	10	53.7	170	2	Tubiquard SCS-F	1457	80
38	220	31.8	70	159	1.01	Tubiquard SCS-F	1783	80
39	220	40	35	161	1.85	Sevophob HFK-F	1521	95
40	190	29.5	40.3	150	1	Tubiquard SCS-F	1596	70
41	220	40	41.8	160.7	1.2	Tubiquard SCS-F	1723	70
42	190	37.5	63.7	170	1.95	Sevophob HFK-F	1711	95

**Table 3 polymers-12-02131-t003:** Verification experiments at optimum conditions of process parameters (A—fabric weight, B—dye concentration, C—hydrophobic concentration, D—temperature, E—pressure and F—the type of the hydrophobic compound) obtained by Design of Experiment for maximized water vapor permeability and maximized resistance to water wetting.

Nr.	A	B	C	D	E	F	Resistance to Water Wetting, % PREDICTED	Resistance to Water Wetting, % EXPERIMENT	Water Vapor Permeability PREDICTED g/m^2^	Water Vapor Permeability EXPERIMENT g/m^2^
1.	190	16.931	41.066	168.903	1.046	Sevophob	102	100	2670	2609
2.	220	56.067	68.802	154.276	1.048	Tubiquard	100	100	2176	2620
3.	240	11.106	38.815	152.460	1.908	Sevophob	88	100	2355	2530

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
