# Peer review of "Design of Experiment Approach to Optimize Hydrophobic Fabric Treatments"

_polymers, 2020, doi:10.3390/polym12092131_

Round 1

Reviewer 1 Report

THE TITLE OF THE ARTICLE DOES NOT MEET THE SCOPE OF THE EXPERIMENT:

  • article is not presenting recent developments (as a review would), but it is presenting a specific scientific experimental work
  • article is not addressing the polymers in general but only fabric, textile. (this issue appears also in the abstract, line 13 and in the text)

Please totally rephrase the title (and elsewhere) to meet the scope of the research.

Line 14: add abbreviation - Design of Experiment (DoE)

Line 23: 154.3 °C and not 154.3°C

Line 23: Unclear statement: It was found … Please rewrite.

Line 24: Inappropriate statement. DoE is not a tool for surface treatment but it is a tool for optimisation of parameters. Please rewrite.

Inappropriate keywords: polymer surface, waterproof fabric. Select different ones (to be more aligned with the scope of the research or used materials)

Line 29 to 43: Introduction starts with antibiotic-resistance. This is not appropriate since THE RESEARCH IS NOT USING ANY METHOD FOR ITS DETERMINATION AND IS JUST INDIRECTLY ASSUMING ON IT by measurements of water vapour permeability and water wetting. Please delete this paragraph or somehow incorporate in the middle of the introduction with explanation of mentioned assumption. For the same reason review all article to avoid any misuse (for example line 104 …).

Line 38, Table 1 and elsewhere: please add a space between the number and the unit (for example: 13 % and not 13%, 13 °C and not 13°C). The only exception is in case of the units for geometry angle (for example 13°).

Line 104: This statement is not properly addressing the highlighted toxicity (line 98 to 103) on which this statement is connecting (and again misuse of the term polymer). Also the aim of the research was different, as it is well presented in lines 351 to 356.

Line 118, 179, 180, 181, 320, 321: When addressing the values from and to, please add the unit also at the first value (for example: from 20 % to 30 %, and not from 20 to 30 %).

Line 126: delete repellency

Line 158: (2.3)? maybe (2.5)

Line 160 to 165: Schematic overview would be useful

Line 172 to 204: this content fits more to the chapter 2 Materials and methods

Line 207 to 212: Only 3 repeated measurements are not enough to properly assume on reproducibility from which we could determine experimental error.

Line 207 to 216: A new title for this section would be useful

Line 241: instead of “next” use number

Line 260 to 264: Could be moved to 2. Materials and Methods and explained a bit more.

Line 267: once it is written “8 C” (or 6 C), next ones is 8C. Please uniform through the whole paper.

Line 275 to 278: reference to a literature is needed

Table 3: units are missing

Line 319: For direct measurement of hydrophobicity other methods are more appropriate. Assessment of water wetting is indirect method. Please review the whole content and in some cases replace hydrophobicity with more appropriate expression.

Line 346 to 350: Already commented. This issue was not experimentally addressed so there is no reason to include this content here.

Line 367 to 369: Unclear conclusion. Please rewrite.

Author Response

Answer to „Comments and Suggestions for Authors from the Reviewer 1“:

Dear Reviewer, thank you very much for your valuable comments that have made this manuscript much improved.

  1. THE TITLE OF THE ARTICLE DOES NOT MEET THE SCOPE OF THE EXPERIMENT: article is not presenting recent developments (as a review would), but it is presenting a specific scientific experimental work, article is not addressing the polymers in general but only fabric, textile. (this issue appears also in the abstract, line 13 and in the text). Please totally rephrase the title (and elsewhere) to meet the scope of the research.

RE: New title is: “Design of Experiment Approach to Optimize Hydrophobic Fabric Treatments”

Line 14: add abbreviation - Design of Experiment (DoE)

RE: DoE Abbreviation is added

Line 23: 154.3 °C and not 154.3°C

RE: This correction is made

Line 23: Unclear statement: It was found … Please rewrite.

RE: Correction is made and unclear statement is rewritten as: Optimal process contained following parameters: 154.3 °C, 1.05 bar, 56.07 g/L dye, 220 g/m2 fabric.

Line 24: Inappropriate statement. DoE is not a tool for surface treatment but it is a tool for optimisation of parameters. Please rewrite.

RE: Correction is made and unclear statement is rewritten as: Therefore it is shown that DoE is an excellent and efficient tool for optimization of parameters used in polymer surface treatment.

Inappropriate keywords: polymer surface, waterproof fabric. Select different ones (to be more aligned with the scope of the research or used materials)

RE: Correction is made and new keywords are chosen

Line 29 to 43: Introduction starts with antibiotic-resistance. This is not appropriate since THE RESEARCH IS NOT USING ANY METHOD FOR ITS DETERMINATION AND IS JUST INDIRECTLY ASSUMING ON IT by measurements of water vapour permeability and water wetting. Please delete this paragraph or somehow incorporate in the middle of the introduction with explanation of mentioned assumption. For the same reason review all article to avoid any misuse (for example line 104 …).

RE: Correction is made and now this paragraph is incorporated in the last part of the introduction (lines 104 – 114).

Line 38, Table 1 and elsewhere: please add a space between the number and the unit (for example: 13 % and not 13%, 13 °C and not 13°C). The only exception is in case of the units for geometry angle (for example 13°).

RE: This correction was made

Line 104: This statement is not properly addressing the highlighted toxicity (line 98 to 103) on which this statement is connecting (and again misuse of the term polymer). Also the aim of the research was different, as it is well presented in lines 351 to 356.

RE: Line 104 is corrected (It is now line 91).

Line 118, 179, 180, 181, 320, 321: When addressing the values from and to, please add the unit also at the first value (for example: from 20 % to 30 %, and not from 20 to 30 %).

RE: This correction was made

Line 126: delete repellency

RE: This correction was made

Line 158: (2.3)? maybe (2.5)

RE: This correction was made

Line 160 to 165: Schematic overview would be useful

RE: This correction was made, Figure 2 now shows the schematic overview of the process performed

Line 172 to 204: this content fits more to the chapter 2 Materials and methods

RE: This correction was made

Line 207 to 212: Only 3 repeated measurements are not enough to properly assume on reproducibility from which we could determine experimental error.

RE: In the Design of Experiment program, the number of repetitions are predifined as three repeated measuruments. This means that we performed three repeated measurements on three repeated points, which makes nine repetitions in total. From this, statistical evaluation was done.

Line 207 to 216: A new title for this section would be useful

RE: The 3.1. Title is changed into the Optimization by Design of experiments

Line 241: instead of “next” use number

RE: This correction was made, (it is now Line 292).

Line 260 to 264: Could be moved to 2. Materials and Methods and explained a bit more.

RE: This correction was made, the paragraph is moved to Materials and Methods (sub section 2.6.) with explanation.

Line 267: once it is written “8 C” (or 6 C), next ones is 8C. Please uniform through the whole paper.

RE: This correction was made, and unified way of writting is used as 8 C or 6 C atoms.

Line 275 to 278: reference to a literature is needed

RE: The references are added (20 and 21). In addition, new reference 28 is added to the manuscript in order to enhance its value

Table 3: units are missing

RE: The references are added to the Table 3

Line 319: For direct measurement of hydrophobicity other methods are more appropriate. Assessment of water wetting is indirect method. Please review the whole content and in some cases replace hydrophobicity with more appropriate expression.

RE: This correction was made, and the term „resistance to water wetting“ was used to replace the term „hydrophobicity

Line 346 to 350: Already commented. This issue was not experimentally addressed so there is no reason to include this content here.

RE: This correction was made, and this paragraph is deleted from the manuscript.

Line 367 to 369: Unclear conclusion. Please rewrite.

RE: This correction was made, and the Conclusion is now partialy rewritten.

Reviewer 2 Report

This manuscript describes a systematic experimental approach to fabricate hydrophobic fabrics using experimental design and annova analyses. The authors use different fabrics and commercial formulation to achieve their goal. They also use an efficient optimization protocol to select the best hydrophobic and barrier fabrics with treatment. In general, the paper is interesting for the journal Polymers however, definitely major revisions as noted below are needed before further due is given for publication:

  1. Please change the title. It reads like the title of a review article. Put a title that describes your work better like “Design of Experiment Approach to Optimize Hydrophobic Fabric Treatments”.
  2. Manuscript English grammar needs a good degree of editing.
  3. Please indicate the chemistry and type of the fluorine tremants like fluorinated wax, or fluorinated acrylic etc.
  4. Please also cite and discuss a recent review on ecofriendly treatment of textiles: Advances in Colloid and Interface Science Volume 270, August 2019, Pages 216-250.
  5. Axes labels of figures 4 to 8 are very small and not visible. Please fix those graphs with better labeling.
  6. Actually, the above problem is valid for almost all your figures. Please enlarge all the fonts.
  7. Please design a better Figure 2 that describes the test. Otherwise remove it from the paper.
  8.  

Author Response

Answeres to Reviewer 2:

RE: Dear Reviewer, thank you for your valuable comments that have made this manuscript much improved

Comments and Suggestions for Authors

  1. Please change the title. into “Design of Experiment Approach to Optimize Hydrophobic Fabric Treatments”.

RE: New title is corrected as is advised by the Reviewer 2, and it is now: “Design of Experiment Approach to Optimize Hydrophobic Fabric Treatments”. The changes are marked in the text with red color.

  1. Manuscript English grammar needs a good degree of editing.

RE: English was revised for gramar and style.

  1. Please indicate the chemistry and type of the fluorine tremants like fluorinated wax, or fluorinated acrylic etc.

RE: The type of the fluorine treatments is now explained in detail, refering to Sevophobe as fluorocarbon resin which is used for permanent water, oil and dirt-repellent finishing, and Tubiquard as low viscous liquid dispersion.

  1. Please also cite and discuss a recent review on ecofriendly treatment of textiles: Advances in Colloid and Interface Science Volume 270, August 2019, Pages 216-250.

RE: New reference [28] is cited and discussed within the manuscript, on page 11:

 “Zahid et al. reported that among materials with extra developed hydrophobic properties textile materials are coming in contact with the human skin most frequently. Therefore the authors have emphasized that textile treatments for water or oil repellency should be non-toxic, biocompatible, and comply with stringent health standards [28]. Moreover, due to large volume of the worldwide used quantitites of water, chemicals and reagents used in the textile industry, treatments should be scalable, sustainable, and eco-friendly. Due to this awareness, new eco-friendly processes are being developed and adopted.

Moreover, the review article of Zahid et al. reported that although fluorinated polymers with C8 chemistry are the best performing materials to render textiles water or oil repellent, they pose substantial health and environmental problems and are being banned. Therefore C8-free vehicles for non-wettable treatment formulations are probably the only ones that can have important commercialization prospects. In addition, their review article indicated promising future strategies and new materials that can transform the process for non-wettable textiles into an all-sustainable technology [28].”

  1. Axes labels of figures 4 to 8 are very small and not visible. Please fix those graphs with better labeling. Actually, the above problem is valid for almost all your figures. Please enlarge all the fonts.

RE: Axes lebels shown in Figures 4 to 8 have default sizes defined by the software program. Nevertheless, new labels are now manually added to all figures where appropriate.

  1. Please design a better Figure 2 that describes the test. Otherwise remove it from the paper.

RE: New schematic overview of process used is now placed instead of the previous Figure 2.

Round 2

Reviewer 2 Report

I have found the revisions adequate and satisfactory. I also declare that I have no further technical or scientific issues with the revised version of the manuscript and am in the opinion that it may be published without further alteration in Polymers.